# Chromosome-level genome assembly of *Bactrocera dorsalis* reveals its adaptation and invasion mechanisms

Fan Jiang[1], Liang Liang[2], Jing Wang[3] & Shuifang Zhu [1,4✉]

*Bactrocera dorsalis* is an invasive polyphagous pest causing considerable ecological and economic damage worldwide. We report a high-quality chromosome-level genome assembly and combine various transcriptome data to explore the molecular mechanisms of its rapid adaptation to new environments. The expansions of the DDE transposase superfamily and key gene families related to environmental adaptation and enrichment of the expanded and unique gene families in metabolism and defence response pathways explain its environmental adaptability. The relatively high but not significantly different expression of heat-shock proteins, regardless of the environmental conditions, suggests an intrinsic mechanism underlying its adaptation to high temperatures. The mitogen-activated protein kinase pathway plays a key role in adaptation to new environments. The prevalence of duplicated genes in its genome explains the diversity in the *B. dorsalis* complex. These findings provide insights into the genetic basis of the invasiveness and diversity of *B. dorsalis*, explaining its rapid adaptation and expansion.

[1] Chinese Academy of Inspection and Quarantine, Beijing 100176, China. [2] Academy of Agricultural Planning and Engineering, MARA, Beijing 100125, China. [3] Biomarker Technologies Corporation, Beijing 101300, China. [4] Sanya Research Institute of Chinese Academy of Tropical Agricultural Sciences, Hainan 572025, China. ✉email: zhusf@caiq.org.cn

The oriental fruit fly, *Bactrocera dorsalis* (Diptera, Tephritidae), is a highly invasive pest. Native to the Indo-Asian region, *B. dorsalis* is found in at least 65 countries in six continents (CABI, https://www.cabi.org). It can easily spread and establish populations after its introduction to a new area, including the areas that were previously considered climatically unsuitable for *B. dorsalis*[1]. Once introduced, it can easily spread because of its high reproductive potential, short life cycle, and broad host range[2]. Its host range includes more than 250 fruit and vegetable crops[3]; therefore, the infestation by fruit fly imposes significant economic losses primarily due to direct fruit damage and the accrued cost for export limitations associated with quarantine and eradication measures[4–6]. Furthermore, *B. dorsalis* is highly competitive with other invasive and harmful tephritid fruit flies and quickly becomes the dominant tephritid species in a new area[7,8]. For instance, within a short time of introduction of *B. dorsalis* in Hawaii, the population of *Ceratitis capitata*, another major tephritid fruit fly pest of economic importance, declined severely, such that adult flies or infested fruits could rarely be found in coastal areas[2]. However, the genetic basis of the adaptability and invasiveness of *B. dorsalis* has not been completely explored, despite its threats to the economy, environment, and biodiversity.

Genomic tools, particularly, the availability of a high-quality assembled genome, facilitate exploring the genetic basis for the global spread and diversity of different organisms[9]. However, despite several research advances in the genomics field in other organisms, in the Tephritidae family, the genomes of only 11 species have been sequenced and assembled with contig N50 from 0.387 to 350.91 kb. The genome annotations of only six species have been released [available at the National Centre for Biotechnology Information (NCBI)], and only a scaffold-level genome assembly of *B. dorsalis* (PRJNA273958) with low-quality (contig N50 of 4.91 kb) has been reported to date. Moreover, a group of nearly 100 taxa sharing morphological and genetic similarities with *B. dorsalis* but different from other tephritid fruit flies form the *B. dorsalis* complex[10]. The invasive potential of members of this complex is similar to that of *B. dorsalis*, possibly owing to the similarity in their genetic characteristics. Therefore, understanding the genetic relatedness of *B. dorsalis* is a prerequisite for developing efficient strategies to prevent and control these pests. Furthermore, combining the transcriptome data or genome resequencing with genome-wide association studies has been proved efficient for understanding the mechanisms of important traits in different organisms[11,12]. Therefore, we hypothesised that elucidation of the genomic features of *B. dorsalis* followed by a comparative analysis with other tephritids and other economically significant insects could help understand the molecular mechanisms underlying the rapid adaptation of *B. dorsalis* to new environments.

To test this hypothesis, this study developed a high-quality chromosome-scale assembly of the *B. dorsalis* genome using the PacBio and Illumina platforms assisted by the high-throughput chromosome conformation capture (Hi-C) technique. The comparative analysis of the genomes of *B. dorsalis* with those of the other tephritids and other economically significant insects explored the genomic features specific to *B. dorsalis*. Furthermore, by combining various transcriptome data, we further investigated the genetic basis underlying its high invasiveness and rapid adaptation to new environments, which could facilitate developing effective prevention and control strategies to reduce damage caused by outbreaks of *B. dorsalis*.

## Results and discussion

### Genome sequencing and chromosome-level genome assembly.

We sequenced the genome of *B. dorsalis* using single-molecule real-time sequencing (SMRT) (PacBio Sequel), paired-end sequencing (Illumina Hiseq), and Hi-C technique (Phase Genomics, Inc.). Using Illumina reads, the genome size of *B. dorsalis* was estimated to be ~522.76 Mb through k-mer analysis, and the estimated heterozygosity was ~2.2% (Supplementary Fig. 1). After quality control and filtering, 30.92 Gb (57 × fold coverage) clean PacBio subreads with a mean read length of 9.391 kb were generated and used to assemble the 538.24 Mb *B. dorsalis* genome with a contig N50 size of 1.06 Mb. After polishing, 83.79 Gb of the clean Hi-C data (155 × fold coverage) were used to correct the assembled genome (Supplementary Tables 1–3). After error correction and validation, the assembly of the *B. dorsalis* genome yielded a 542.04 Mb reference genomic sequence with a contig N50 size of 1.12 Mb. Although the whole genome sequences of *C. capitata*[13] and *B. cucurbitae*[14], which also belong to the Tephritidae family, have been previously reported, here, we report the high-quality assembled genome of *B. dorsalis*, which was larger than the genomes of the other sequenced tephritids (Supplementary Table 4).

According to the Hi-C interaction information, unique, valid interaction pairs were mapped onto the draft assembly contigs and divided into six chromosomes using the LACHESIS software (Fig. 1 and Supplementary Fig. 2); the connecting lines in the centre of the diagram show homologous relationships of chromosomes and translocated regions. The genome of *B. dorsalis* had a high degree of completeness (99.78 and 96.96%) when compared with the Eukaryotic data set of Core Eukaryotic Genes Mapping Approach (CEGMA v2.5)[15] and the Arthropoda data set of the Benchmark of Universal Single-Copy Orthologs (BUSCO v2.0)[16] (Supplementary Tables 5, 6).

### Genome annotation.

A total of 15,775 genes were obtained via ab initio and homologous prediction after removing repeat sequences; 97.51% of the genes were functionally annotated using the Non-Redundant Protein Sequence Database (NR), Eukaryotic Orthologous Groups (KOG), Kyoto Encyclopedia of Genes and Genomes (KEGG), and TrEMBL databases (Supplementary Table 7). Gene Ontology (GO) assignments were used to predict the functions of *B. dorsalis* genes by classifying them into three categories: biological process, molecular function, and cellular component[17]. The analysis identified 3,434 biological process, 720 molecular function, and 1900 cellular component GO terms. The enriched GO categories included cellular and single-organism processes, organelle and membrane, transcription factor, and protein-binding activity. We also used KEGG[18] to screen all genes for pathway annotation using Fisher's exact test ($P < 0.0001$). In total, 4,480 genes were annotated, and 162 pathways, including purine metabolism, ribosome, and spliceosome pathways, were enriched. In addition, we obtained 250.36 Mb repeat sequences, 493 tRNA, 32 rRNA, 59 miRNA, and 1,393 pseudogenes and predicted 1600 motifs (Supplementary Tables 8, 9).

Of the predicted *B. dorsalis* genes, 2,186 (13.86%) genes were present as tandem duplicates and formed 1,345 duplication events, 233 (1.48%) were segmentally duplicated genes and formed 143 duplication events. The comparative analysis of the tandemly and segmentally duplicated genes in *B. dorsalis* and 13 other economically significant insects (*B. latifrons*, *B. oleae*, *Zeugodacus cucurbitae*, *C. capitata*, *Rhagoletis zephyria*, *Drosophila melanogaster*, *Lucilia cuprina*, *Musca domestica*, *Bombyx mori*, *Spodoptera litura*, *Danaus plexippus*, *Plutella xylostella*, and *Locusta migratoria*) demonstrated a high prevalence of these genes in *B. dorsalis* next to *D. plexippus* ($n = 2,460$; 16.26%) and *B. mori* ($n = 320$; 1.90%), respectively (Supplementary Table 10). These results indicated that compared to other tephritids and dipterans, the duplicated genes are prevalent in *B. dorsalis*. It is

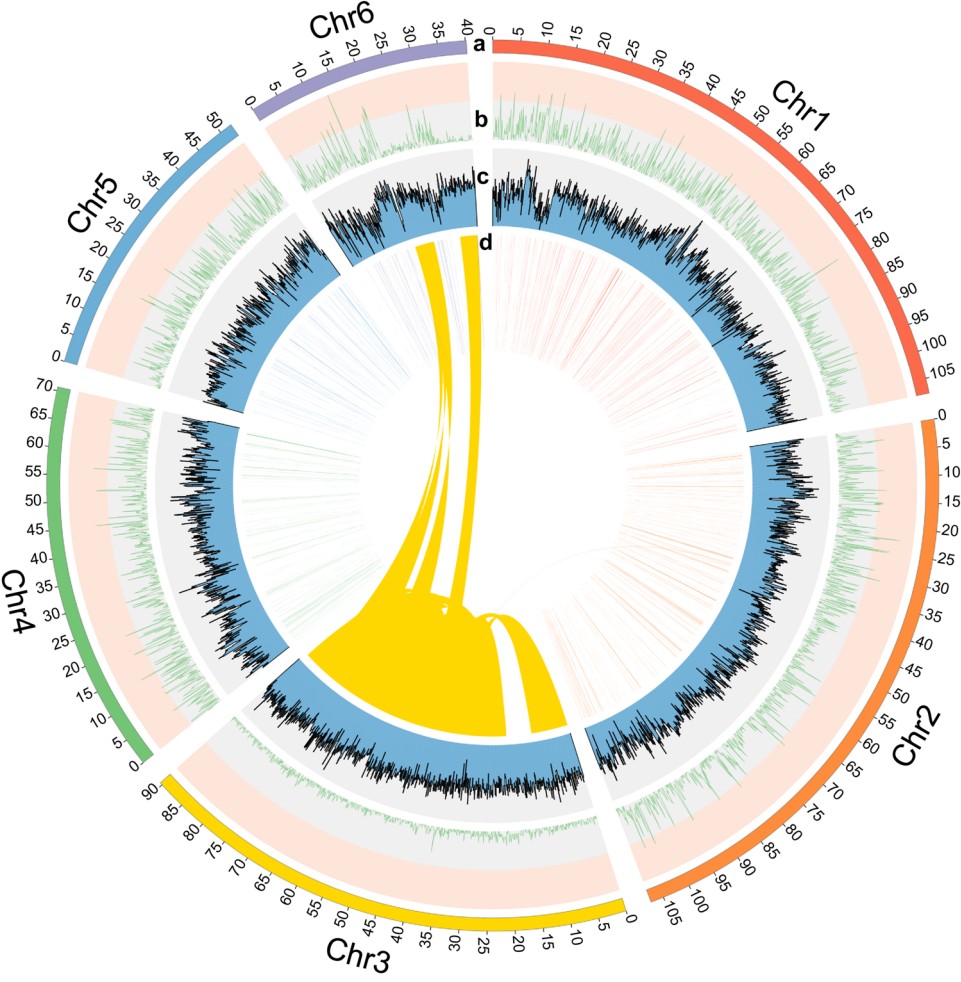

**Fig. 1 Circular diagram depicting the characteristics of the *Bactrocera dorsalis* genome.** From the outer to the inner circle: **a** Ideogram of the six *B. dorsalis* chromosomes at the Mb scale. **b** Gene density. **c** Guanine-cytosine (GC) content. **d** Collinearity block in the genome. Connecting lines at the centre of the diagram highlight the homologous relationships of chromosomes.

known that duplication contributes to species differentiation[19]. In other words, these results explained the species diversity in the *B. dorsalis* complex.

Among the segmentally duplicated genes, most syntenic genes encoded endonucleases of the DDE superfamily, a family of transposase proteins necessary for efficient DNA transposition. Compared with 13 other species, the number of DDEs in *B. dorsalis* was the second highest ($n = 121$; 0.77%) after *R. zephyria* ($n = 402$, 1.58%); however, the number of segmentally duplicated genes that comprised the DDE family genes were the highest in *B. dorsalis* ($n = 40$) (Supplementary Table 10). The DDE family genes act via DNA cleavage at a specific site, followed by a strand transfer reaction[20]. Furthermore, the function of the DDE superfamily is likely related to rearrangement, homologous and non-homologous DNA editing, and integration and removal of invaders (such as pathogens)[21]. Reportedly genetic novelty is associated with adaptive evolution[22], and transposition, foreign pathogens, and nucleic acid manipulation are important factors for evolution in insects[21]. Our results showed that the drastic expansion of the DDE superfamily in *B. dorsalis* potentially contributed to the pressure of adaptation to different environments. Furthermore, chromosome 6 shared high synteny with chromosome 3. These two chromosomes demonstrated frequent segmental duplication and retained numerous duplicated chromosomal blocks, suggesting that genes in these two chromosomes might have been translocated via chromosome rearrangements.

**Genome evolution**. The gene families among *B. dorsalis* and other 13 economically important insects were identified using Orthofinder v2.3.7 (Supplementary Tables 11, 12). Our results showed that the composition of the *Bactrocera* proteome is relatively common. To determine the evolutionary status of *B. dorsalis*, a maximum-likelihood phylogenetic tree was constructed based on 786 single-copy genes shared by *B. dorsalis* and the above-mentioned species and the MCMCtree programme was used to estimate divergence among these species. The results revealed that *B. dorsalis* clustered with *B. latifrons* in the *Bactrocera* genus, and the divergence between these species was estimated to have happened 3–9 million years ago (Mya) (Fig. 2 and Supplementary Fig. 3). Furthermore, *B. dorsalis* and *C. capitata* shared a last common ancestor, approximately 30–64 Mya (Fig. 2 and Supplementary Fig. 3), indicating *B. dorsalis* is a new species derived from Tephritidae evolution. Compared with *B. latifrons*, *B. oleae*, *Z. cucurbitae*, and *C. capitata*, *B. dorsalis* contained more genes and gene families (Supplementary Table 11 and Supplementary Fig. 4), for example, 73 unique gene families were detected for *B. dorsalis*, while only 12, 48, 21, and 23 unique gene families were identified in *B. latifrons*, *B. oleae*, *Z. cucurbitae*, and *C. capitata*, respectively, indicating that the genes in *B. dorsalis* might have accumulated more functional mutations than their progenitors. In terms of evolution, "key innovation" hypotheses show that evolutionary success is often attributed to key evolutionary innovations or adaptive breakthroughs, and the

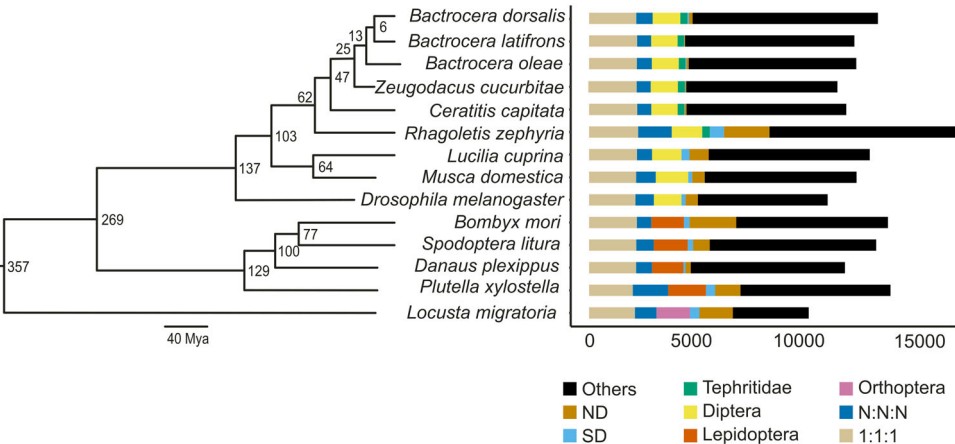

**Fig. 2 Orthology and genome evolution of *Bactrocera dorsalis* compared with those of 13 other species.** The 13 species used for comparison were *B. latifrons, B. oleae, Zeugodacus cucurbitae, Ceratitis capitata, Rhagoletis zephyria, Drosophila melanogaster, Lucilia cuprina, Musca domestica, Bombyx mori, Spodoptera litura, Danaus plexippus, Plutella xylostella,* and *Locusta migratoria.* IQ-TREE was used to construct the unrooted maximum-likelihood phylogenetic tree for the 14 species based on genomic data obtained using LG + F + I + G4 model for 1000 bootstrap replicates. The gene set of each species was subdivided into different types of orthology clusters. '1:1:1' represents universal single-copy gene families across all examined species; 'N:N:N' indicates other universal genes; 'Orthoptera,' 'Lepidoptera,' 'Diptera,' and 'Tephritidae' indicate common gene families unique to Orthoptera, Lepidoptera, Diptera, and Tephritidae; 'S.D.' represents species-specific duplication; 'N.D.' represents species-specific genes.

evolutionary changes that increase individual fitness should be common under selection pressures[23]. "Increased Fitness," one of the key innovation hypotheses that describe increasing competitive ability through evolution, might explain the highly competitive nature of *B. dorsalis* compared with other invasive and harmful fruit flies.

In our assembly, 15,775 *B. dorsalis* genes were clustered into 11,448 gene families. Gene family analysis also revealed that 2,536 *B. dorsalis* gene families were shared with 13 other species (Supplementary Fig. 4), whereas 73 *B. dorsalis* gene families containing 171 genes were unique to *B. dorsalis*. To assess the function of these unique genes, we performed a GO enrichment analysis using clusterProfile v3.14.0[24], which identified several genes involved in stimulus-response and metabolism (Supplementary Fig. 5).

We also examined the expansion and contraction of gene families in *B. dorsalis* and identified 154 and 82 gene families that were expanded and contracted, respectively (Supplementary Fig. 6). The families that expanded significantly were involved in metabolism and defence responses, such as DNA metabolism, innate immune response, and defence response to bacteria (Supplementary Fig. 7), whereas the contracted gene families were enriched in the antimicrobial humoral response (Supplementary Fig. 8). These findings suggest that the expanded and unique gene families of *B. dorsalis* might be closely associated with its enhanced environmental adaptability.

**Gene families associated with adaptation and invasiveness.** We identified some key gene families, including those encoding heat shock proteins (Hsps), mitogen-activated protein kinases (MAPKs), chemosensory receptors, and cytochrome P450 monooxygenase (CYP450s) associated with environmental adaptation in seven insect species, including five tephritids and two ubiquitous species (*D. melanogaster* and *L. migratoria)* with strong adaptability.

**Hsps**. Hsps are found in every organism and are considered responsible for developing tolerance to thermal and various other abiotic stresses[25–27]. The Hsp family was expanded in *B. dorsalis* compared to other tephritid insects. The *B. dorsalis* genome was found to harbour 80 Hsps [34 Hsp40s, 9 Hsp60s, 17 Hsp70s, 4

Hsp90s, and 16 small Hsps (sHsps)], a number similar to that in the *D. melanogaster* genome (82; the highest in the seven species studied). This high number of Hsps in *B. dorsalis* was mainly attributed to the expansion of *Hsp70* and *sHsp*. Compared to other tephritid insects, *B. dorsalis* had the highest number of *Hsp60, Hsp70, Hsp90,* and *sHsp* genes, indicating the expansion of *Hsp* coding genes in Tephritidae (Fig. 3a). It has been reported that the life of a *B. dorsalis* adult is approximately one year in cool mountainous locations and is greatly reduced when the daily maximum temperature is above 40.5 °C[2]. Based on these, it can be inferred that the expansion of the Hsp family especially, Hsp70s and sHsps, might be a general strategy for adaptation to environmental cold and heat stresses in *B. dorsalis*.

Furthermore, gene expression provides clues for the thermal adaptation of *B. dorsalis*. For example, several studies have shown that the upregulation of Hsps is a general strategy for organisms to respond to adverse environmental stresses[28–32]. The comparative transcriptome analyses demonstrated that the median fragments per kilobase of transcript per million mapped (FKPM) reads for Hsps were higher than those of all transcripts in different environmental conditions—3.07 and 1.68, at 25 °C; 2.94 and 1.64 at 38 °C; 3.90 and 2.86, under fed conditions; and 4.06 and 2.80, under starved conditions, respectively (Fig. 3b). However, the expression levels of Hsps at different environmental conditions (25 °C vs 38 °C and fed vs starved) were not significantly different. Furthermore, a previous study has shown that the species with a reduced expression of Hsps were less tolerant to heat stress, which occurred only in stable environments, and the species with high Hsps expression survived after severe heat shock, and their population grew[33]. However, in *B. dorsalis*, the intrinsically high Hsp expression levels and no distinct response of Hsps expression at 38 °C revealed its intrinsic adaptation system to high temperatures. For insects, the temperature is certainly the most critical abiotic factor affecting physiology, wherein the extreme temperatures significantly affect population dynamics[34]. Furthermore, it has been shown that for some invasive insect pests, a rapid adaptation of thermal traits may facilitate survival in novel environments[35]. Taken together, we inferred that the adaptation to extreme temperatures might be a key reason for the high invasiveness of *B. dorsalis*.

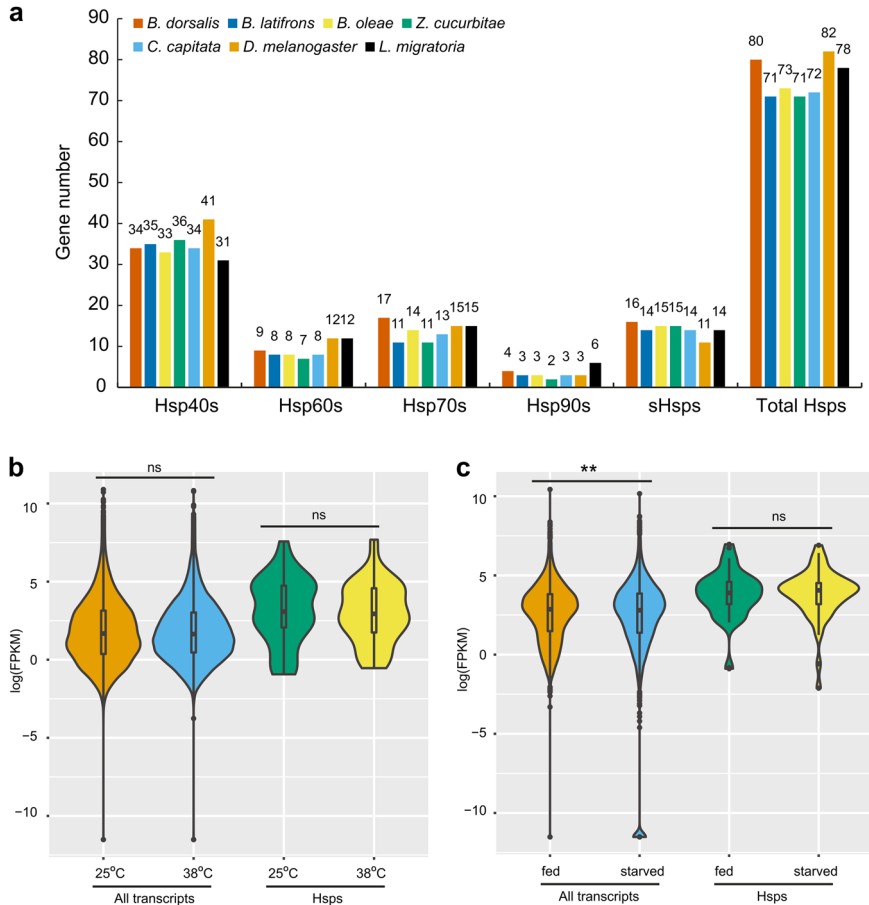

**Fig. 3 Genome and transcriptome analyses of the Hsp gene family in *Bactrocera dorsalis*. a** Comparison of gene numbers for Hsp40, Hsp60, Hsp70, Hsp90, sHsps, and total Hsps in *B. dorsalis*, *B. latifrons*, *B. oleae*, *Z. cucurbitae*, *C. capitata*, *D. melanogaster*, and *L. migratoria*. **b** Comparison of all transcripts and Hsp transcripts under high-temperature (25 vs 38 °C). **c** Comparison of all transcripts and Hsp transcripts under starvation conditions (fed vs starved).

**p38 MAPK pathway**. MAPK pathways are important regulators of cellular responses to various extracellular stimuli. The p38 MAPK pathway is conserved from yeast to mammals, and the p38 MAPK signalling is activated by most environmental stress stimuli, regulating various physiological processes, such as cell differentiation, cell cycle, and inflammation[36]. We found that all key components of the p38 MAPK pathway were well represented in the *B. dorsalis* genome (Fig. 4). In *B. dorsalis*, the MAPK kinase MAP2K3 is the upstream activator of p38 MAPK, responsible for phosphorylation. MAP2K3 is activated by various MAPK kinase kinases, including MEKK4, MAP3K5, MAP3K7, and MAP3K9. Under heat stress, heat shock upregulates the expression of MEKK4 and MAP3K5, which transmit the heat shock signal to MAP2K3, and subsequently activate p38 MAPK. Additionally, p38 MAPK increases the phosphorylation of activating transcription factor 2 (ATF-2), which promotes the expression of dual oxidase (Duox) and results in the generation of reactive oxygen species, leading to an increase in oxidative stress. In addition, p38 MAPK modulates the expression of the mitochondrial sodA enzyme through phosphorylation of the transcription factor MEF2, thus leading to changes in longevity[37,38]. Here, we demonstrate the upregulation of multiple components of the p38 signalling pathway and downregulation of MAP2K3 under heat stress in *B. dorsalis*. Taken together, we conjectured that the *B. dorsalis* p38 MAPK pathway could respond to heat stress, and MAP2K3 might be a negative regulatory factor to prevent the excessive activation of the p38 signalling pathway.

The p38 MAPK pathway is also important under various abiotic stress conditions (e.g., low temperature, oxidative stress, UV light) in other insects[36,39]. It plays a crucial role in resisting microbial infection in the insect gut. For example, it is the main innate immune response signalling pathway in dipterans *Aedes aegypti* and *D. melanogaster*[40]. The Duox-regulating pathway regulates the Duox expression through p38 MAPK to respond to pathogenic microorganisms, avoid massive immune responses against symbiotic microbes, and maintain homeostasis in the gut[40]. The larvae of *B. dorsalis* generally live in rotten fruits and have numerous opportunities to be exposed to pathogens. Thus, they have to rely on their innate immune system to combat infecting microbes, probably through p38 MAPK. Furthermore, the gut microbiota is also pivotal in increasing the tolerance to low-temperature stress in *B. dorsalis*[35]. Based on these results, we infer that the p38 MAPK pathway is essential for the response of *B. dorsalis* to the changing environmental conditions and might be a central homeostatic mechanism under abiotic stress in *B. dorsalis*. Overall, the findings demonstrate that the key components in the p38 MAPK pathway could be potential molecular targets for controlling *B. dorsalis*. However, it is necessary to investigate how upstream and downstream genes are regulated by the p38 MAPK signalling pathway under various abiotic stresses and clarify the regulatory mechanisms involved in improving the anti-stress activity in *B. dorsalis*.

**Chemoreception**. The chemosensory system mediates insect behaviours, such as the location of food, oviposition sites, and

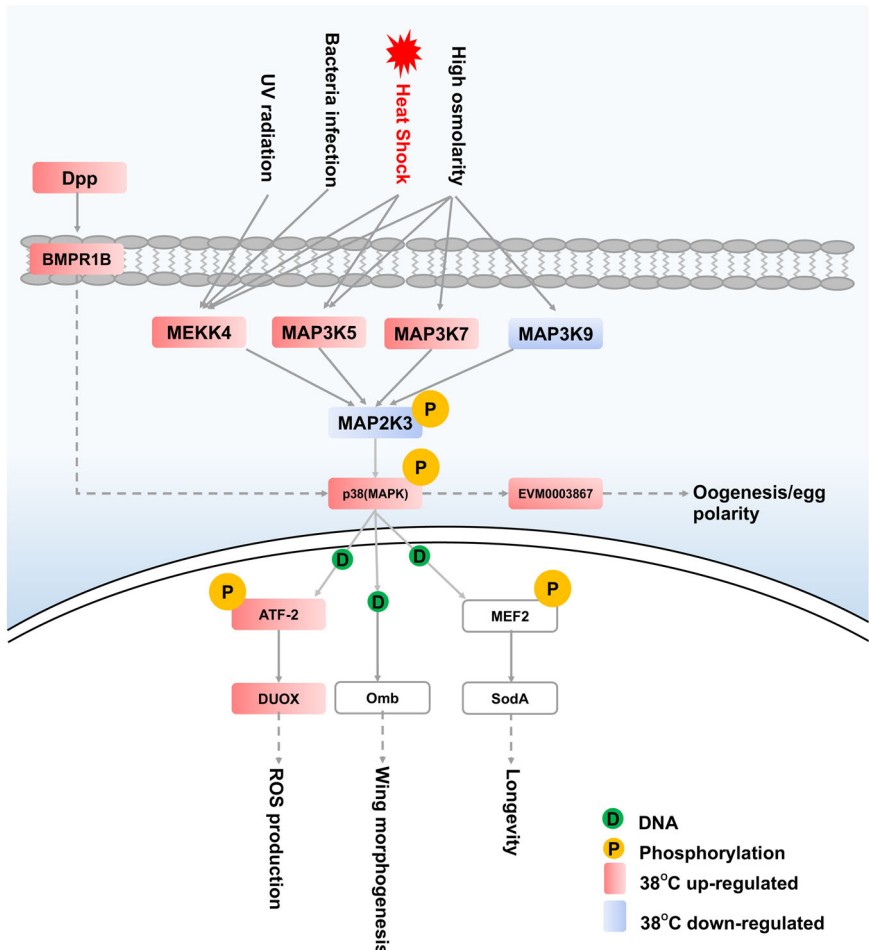

**Fig. 4 P38 MAPK pathway in *Bactrocera dorsalis*.** The activation of p38 mitogen-activated protein kinase regulates a variety of processes through various signalling pathways in *B. dorsalis*. The circles with 'D' and 'P' indicate DNA and phosphorylation, respectively, while the red and blue arrows represent up- and downregulation after treatment at 38 °C.

mates[41]. *B. dorsalis* detects appropriate food sources to maintain a high reproduction rate (a female can lay up to 3062 eggs in a lifetime and 136 per day; under common field conditions, females lay between 1200 and 1500 eggs in a lifetime) through chemoreception[42]. Chemoreception proteins are mainly members of chemoreception-related gene families, such as odorant-binding proteins (OBPs), gustatory receptors (GRs), olfactory receptors (ORs), and ionotropic receptors (IRs)[43–45]. Overall, the *B. dorsalis* genome encodes a similar repertoire of chemoreception-related genes to other tephritid insects and *D. melanogaster*. We identified 52 components in the *B. dorsalis* GR family, representing the greatest expansion of GRs in a tephritid species (Fig. 5a). A notable exception was that a total of 73 ORs were identified, representing a massive gene expansion in the *B. dorsalis* genome (Fig. 5a). Some *B. dorsalis* OR genes orthologous to *Z. cucurbitae* OR7a in the OR-VII group were *B. dorsalis*-specific and clustered, and eight such genes showed a *B. dorsalis*-specific expansion (Fig. 5b). Previous studies showed that gene repertoires belonging to the OR7a family are necessary to detect semiochemicals commonly recognised by fruit flies[46,47]. The expansion of gene families plays a vital role in the evolutionary adaptation of an organism to the environment[48], and it has been shown that chemosensory receptors, particularly OR genes, have evolved under positive selection in insects[49]. Based on these findings, we propose that the expansion of ORs could help *B. dorsalis* decide its behaviours crucial for population establishment and the rapid spread of an invasive species more easily. In addition,

chemoreception and metabolic systems provide useful information to understand how organisms adapt to environments due to the connection between external environmental signals and internal physiological responses[50,51]. Therefore, considering the expansion of the specific genes and gene families enriched in metabolism and defence responses, we infer that these expanded gene families are likely to be particularly advantageous for invasive adaptation of *B. dorsalis*, probably by improving their capacity for obtaining energy, utilising nutrients, reproduction, and stress resistance.

**Detoxification.** Detoxification systems are important for insects to resist the effects of numerous toxins[51]. Resistance to insecticides is considered a major challenge for the integrated pest management of *B. dorsalis*[52]. Cytochrome P450s (CYP450s) and glutathione S-transferases (GSTs) form two major detoxification enzyme families. In this study, we identified 102 CYP450s and 34 GSTs in *B. dorsalis*, which were higher than those in *C. capitata* (98 P450s and 28 GSTs) (Supplementary Table 13). CYP450s have two major functions: synthesis of hormones, which are important to insect development and reproduction, and chemical metabolism, which promotes host adaptation and survival in toxic environments, such as insecticide detoxification[53,54]. A previous study has shown that the number of detoxification genes is related to host usage in insects[55]. These findings suggest that the insecticide resistance and survivability of *B. dorsalis* in

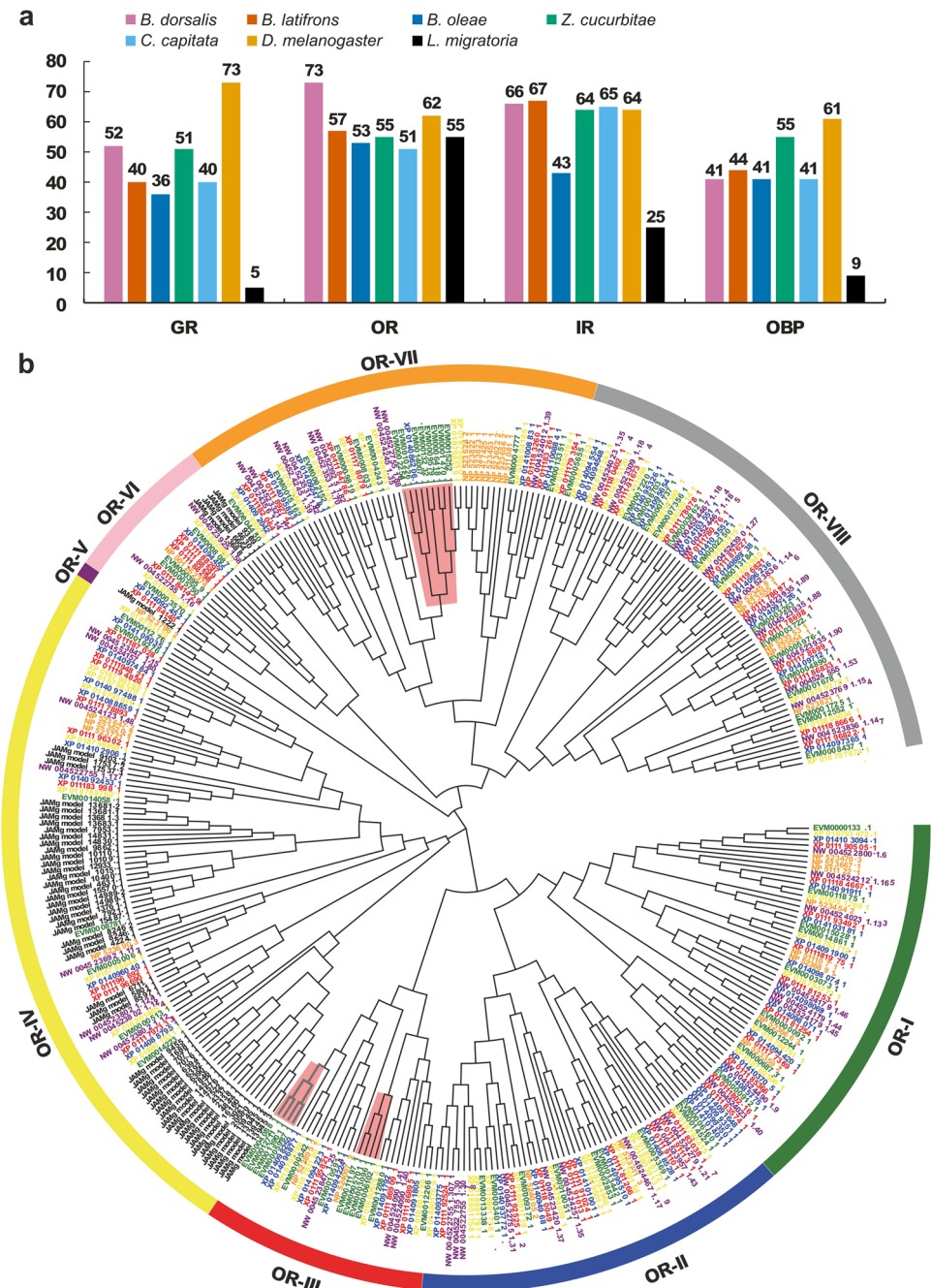

**Fig. 5 Gene families involved in chemoreception in *Bactrocera dorsalis*. a** Comparison of gene numbers for GR, OR, IR and OBP proteins in *B. dorsalis*, *B. latifrons*, *B. oleae*, *Z. cucurbitae*, *C. capitata*, *D. melanogaster*, and *L. migratoria*. Gene numbers are provided above each bar. **b** Phylogenetic relationships of OR proteins from *B. dorsalis*, *B. latifrons*, *B. oleae*, *Z. cucurbitae*, *C. capitata*, *D. melanogaster*, and *L. migratoria*. IQ-TREE was used to build the unrooted maximum-likelihood phylogenetic tree with the LG + G4 model for 1000 bootstrap replicates. OBPs odorant-binding proteins; GRs gustatory receptors; ORs olfactory receptors; IRs ionotropic receptors.

extreme environments could be attributed to the expansions of the P450 and GST detoxification enzyme families.

## Conclusion

Here, we report a high-quality chromosome-level genome assembly of *B. dorsalis*. The *B. dorsalis* genome (542.04 Mb), with a contig N50 size of 1.12 Mb, is anchored into six chromosomes. Enrichment analyses of the expanded and unique gene families in the *B. dorsalis* genome suggest the evolutionary adaptation of *B. dorsalis*. The possible roles of the expanded genes (DDE, Hsp, chemoreception, and detoxification genes) and the p38 MAPK

pathway in its adaptation to environmental stresses are evident from the findings. Furthermore, the diversity of the species comprising the *B. dorsalis* complex is explained by the prevalent duplicated genes in the *B. dorsalis* genome compared with other Diptera species. Overall, the chromosome-level genome assembly constructed herein significantly improves our understanding of *B. dorsalis* genetics and would facilitate further research to gain insights into its population structure using whole-genome resequencing and comparative studies of Tephritidae evolution. Moreover, the outcomes of this study are of high scientific and theoretical significance to comprehensively understand the

mechanism of pest tolerance to stresses and develop innovative pest prevention and control strategies. Based on our findings, the functional roles of the key genes and biological pathways associated with the adaptation and invasiveness of *B. dorsalis* should be focused on in the future to identify the target genes for the precise control of *B. dorsalis*.

## Methods

**Fruit fly samples**. The *B. dorsalis* strain was derived from inbred laboratory strains, which were produced through more than 60 generations of sib mating at the Chinese Academy of Inspection and Quarantine (CAIQ), China. To further reduce sequence polymorphisms and achieve a high-quality genome, the samples of *B. dorsalis* used for de novo sequencing were obtained from one female body using a single mating pair of this strain. Flies were starved after samples were obtained. Only the thorax of each sample was retained for genome sequencing. The eggs, larvae, pupae, and mixed *B. dorsalis* male and female adults were used for transcriptome sequencing using an Illumina HiSeq2500 platform with paired-end libraries for subsequent genome annotation.

**Genome sequencing**. Genomic DNA was extracted via the cetyltrimethylammonium bromide (CTAB) method. Briefly, beta-mercaptoethanol was added to the CTAB extraction buffer (0.1 M Tris-HCl, 0.02 M EDTA, 1.4 M NaCl, 3% (w/v) CTAB and 5% (w/v) PVP K40). DNA was purified and concentrated after being sheared by a G-Tube (Covaris) with 20 kb settings for sequencing. We used the PacBio Sequel sequencing platform by Biomarker Technologies Co., Ltd. to achieve deep coverage of sequencing reads. Libraries were constructed as previously described for SMRT sequencing[56]. Precisely 31.45 Gb of PacBio sequencing data were generated, and 3,292,878 filtered polymerase reads with a Read N50 of 14,289 bp, and an average length of 9,391 bp were yielded (Supplementary Table 14). PacBio subreads ≥ 500 bp were obtained for *B. dorsalis* genome assembly.

To improve the contiguity and ensure the accuracy of assembly results, a 270 bp insert fragment was used to construct sequencing libraries according to the manufacturer's instructions (Illumina HiSeq2500).

To improve draft genome assemblies and create chromosome-length scaffolds, Hi-C technology was also used in this study[57]. Hi-C libraries were created from *B. dorsalis* whole-blood cells, as described in a previous study[58]. For this purpose, the larvae of *B. dorsalis* obtained before death were used. Finally, the Hi-C libraries were paired-end sequenced[59]. A total of 83.79 Gb clean Hi-C reads were aligned to evaluate the ratio of mapped reads, distribution of insert fragments, sequencing coverage, and the number of valid interaction pairs (Supplementary Table 14)[60]. Unique mapped reads spanning two digested fragments comprising distally located but physically associated DNA molecules were defined as valid interaction pairs.

**Genome assembly**. Clean data were assembled using the following steps: (1) longer reads were selected as seed data; (2) supported bases were removed via trimming and hairpin adapters using default parameters with Canu v1.5[61], the longest supported range of error-corrected reads was obtained and then assembled using WTDBG v1.2.8 (https://github.com/ruanjue/wtdbg) with the parameters '-t 64 -H -k 21 -S 1.02 -e 3'; (3) the best draft assembly result was polished. The next-generation sequencing data were used for another correction based on an adopted Pilon algorithm with the parameters '–mindepth 10–changes–threads 4–fix bases'[62–65].

We further clustered and extended the PacBio contigs into pseudochromosomes using Hi-C data. First, a total of 83.79 Gb Hi-C clean reads (155-fold coverage of the *B. dorsalis* genome) were truncated at the putative Hi-C junctions; then, the trimmed reads and single-molecule sequencing contigs were aligned using BWA v0.7.1[66]. We then used LACHESIS[59] to cluster, order, and orient the unique, valid interaction pairs onto chromosomes. Finally, we used PBjelly2 (https://sourceforge.net/projects/pb-jelly/)[67] to fill the corrected SMRT subreads in the gaps. Chromosomes, gene density, GC content, and synteny blocks across the genome were drawn using Circos v0.69-9[68].

**Genome annotation**. For protein-coding prediction and assessment, three gene annotation strategies were used: ab initio prediction using Genscan, Augustus v2.4, GlimmerHMM v3.0.4, GeneID v1.4, and SNAP[69–73]; homologous species prediction using GeMoMa v1.3.1[74]; unigene prediction based on the assembly of transcriptome data from the reference genome using PASA v2.0.2[75]. Finally, EVM v1.1.1 was used to integrate these prediction results with prediction 'Mode:STRANDARD S-ratio:1.13 score>1000', which were modified using PASAv2.0.2[75,76].

For non-coding RNAs, the Rfam and miRBase database were used to predict rRNAs and microRNAs using Blast with e-value 1e−10 and identity cutoff at no less than 95% and INFERNAL v1.1 with the cutoff score at 30 or more, and tRNAscan-SE v1.3.1 was used to predict tRNAs with default parameters[77–80].

For repeat sequence annotation, LTR_FINDER v1.05, MITE-Hunter, RepeatScout v1.0.5, and PILER-DF v2.4 were used to construct a primary repeat sequence database based on structural prediction and the ab initio predication

theory[81–84]. Then, we used PASTEClassifier to classify the primary database and combined this database with the Repbase database to construct the final repeat sequences database. Finally, we used RepeatMasker v4.0.6 for annotation with parameters '-nolow -no_is -norna -engine wublast -qq -frag 20000'[85–87]. Tandem and segment duplicates were detected by MCScanX[88] using the command 'detect_collinear_tandem_arrays.'

For pseudogene prediction, we first predicted candidate pseudogene loci using GenBlastA v1.0.4 (e-value:1e−5) to scan the whole genome for sequences homologous to genes. Then we used GeneWise v2.4.1 to finally determine pseudogenes with premature stop codons and frameshift mutations with parameters '-both -pseudo'[89,90].

For pathway and GO annotation, the predicted genes were used to BLAST against the NR, KOG, KEGG, and TrEMBL databases, with an e-value <1e−5[91–95]. BLAST2GO was used to assign GO terms[96,97]. Motif prediction was performed using the PROSITE, HAMAP, Pfam, PRINTS, ProDom, SMART, TIGRFAMs, PIRSF, SUPERFAMILY, CATH-Gene3D, and PANTHER databases using InterProScanv5.8-49.0[98–109].

**Comparative analyses**. We performed comparative analyses between the genomes of *B. dorsalis* and 13 other representative species, including *B. latifrons*, *B. oleae*, *Z. cucurbitae*, *C. capitata*, *R. zephyria*, *D. melanogaster*, *L. cuprina*, *M. domestica*, *B. mori*, *S. litura*, *D. plexippus*, *P. xylostella*, and *L. migratoria* to explore the unique gene families in *B. dorsalis* and its whole protein-coding gene sets. We constructed a global gene family classification using all-vs-all BLASTP (e-value = 0.001) with default parameters. All-against-all comparison results were clustered using Orthofinder v2.3.7[110]. Species-specific gene families were analysed using the PANTHER v15 database[111] with Fisher's exact test (*P* < 0.0001).

The evolutionary relationships between *B. dorsalis* and the other 13 species were constructed using IQ-TREE[112] with the LG + F + I + G4 model for 1000 bootstrap replicates based on 786 single-copy orthologous protein sequences across all examined species. The divergence between *B. dorsalis* and the other 13 species was estimated using the MCMCtree programme[113] implemented using PAML package v4.9[114] with parameters 'burn-in=10000, sample-number=100000, sample-frequency=2'. Diverge timescale was queried from Timetree (http://www.timetree.org). CAFE v2.2[115] was used to identify the contracted and expanded gene family with 'lambda -l 0.002' and a significance level of *P* < 0.05. KEGG and GO annotation of gene families was performed using BLAST against the NR, KOG, KEGG, and TrEMBL databases with an e-value < 1e−5.

**Gene families**. To understand the invasive and adaptive mechanisms of *B. dorsalis*, we performed comparative analyses of key gene families associated with environmental adaptation in the genomes of seven species, including five tephritids and two ubiquitous species with strong adaptability—*D. melanogaster* and *L. migratoria*.

For chemoreception genes encoding ORs, GRs, IRs, and OBPs, as well as other gene families, including MAPKs and GSTs, we first downloaded the reference protein sequences of some of the tephritids from NCBI GenBank and used BLASTP for the genome assembly of *B. dorsalis* and other six species, with an e-value <1e−5. Then, we used each gene family member to build a Hidden Markov Model (HMM) using HMMER 3.0 (http://www.hmmer.org/) and hmmsearch for the genome assembly of *B. dorsalis* and six other species, with an e-value <1e−5. Finally, the putative gene families were obtained.

For Hsps, including Hsp40, Hsp60, Hsp70, Hsp90, and sHsps, as well as CYP450s, we first downloaded the reference protein sequences of *B. dorsalis*, *D. melanogaster*, and *L. migratoria* from NCBI GenBank then manually built an HMM to search the genome assembly of *B. dorsalis* and six other species with an e-value <1e−5. Second, we used the HMM profiles of Hsps (Hsp40, PF00226; Hsp60, PF00118; Hsp70, PF00012; Hsp90, PF00183; sHsps: PF00011) and P450s (PF00067), which were downloaded from Pfam (http://pfam.xfam.org/), to search against the sequences obtained in the first step, with a threshold e-value <1e−5. Finally, the putative gene families were obtained.

For phylogenetic analysis, we first aligned the protein sequences encoded by each gene family using ClustalW[116] with default parameters; then, we used IQ-TREE[113] to construct the phylogenetic tree with the LG + G4 model for 1000 bootstrap replicates. The phylogenetic trees were visualised using Evolview v3[117].

**Gene expression**. Transcriptome data with accession numbers SRP158095[32] and SRP141127 were downloaded from the NCBI SRA database (https://www.ncbi.nlm.nih.gov/sra/). After removing adapters and filtering low-quality bases ($Q_{phred} < =20$) or N bases (below quality 3) of transcriptome raw reads using TRIMMOMATIC v0.38[118] with parameter 'SLIDINGWINDOW:4:15' and minimum read length of 30 bp. We used HISAT2 v2.1.0[119] with default parameter and STRINGTIE v2.0[120] with 'stringtie <aligned_reads.bam>' to obtain transcripts by mapping clean reads to the *B. dorsalis* genome assembly; the independent sequenced samples were mapped to the *B. dorsalis* genome using BOWTIE2 end-to-end algorithm[121]. For Hsp and MAPK expression, the gene expression level (FPKM) was calculated using RSEM v1.3.1 with default parameters[122].

**Reporting summary**. Further information on research design is available in the Nature Research Reporting Summary linked to this article.

## Data availability

All genome sequence data are available at the GenBank under the Accession number JABETM000000000. Raw sequence data are available at the NCBI SRA site with the accession numbers SRR15444039, SRR15254972, SRR19134783 and SRR19134809.

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

## Acknowledgements
The work was supported by the National Key Research and Development Programme of China (2021YFF0601901) and Basic Scientific Research Foundation of the CAIQ (2018JK008).

## Author contributions
S.F.Z. and F.J. designed the project. F.J. and L.L. collected samples. F.J. and J.W. performed analyses. F.J. wrote the manuscript. S.F.Z. and L.L. improved the manuscript.

## Competing interests
The authors declare no competing interests.
