## [Peer Review File · Communications Biology]

Reviewers' comments:

Reviewer #1 (Remarks to the Author):

Review of "Chromosome-level genome assembly of *Bactrocera dorsalis* reveals adaptation and invasion mechanisms" by Fan Jiang and colleagues.

The authors present their work on *Bactrocera dorsalis*, which is a very problematic pest that adversely affects agriculture in many countries around the world would like very much to have better control methods for. The importance of the subject is very high. It is overall a nice piece of work in Tephritidae comparative evolution that will be of interest to many, due to both major agricultural importance and the closeness of the species to the *Drosophila* community. I think this will be an example of the importance of gene content change analysis.

First the authors present a high quality chromosome scale reference assembly, generated using a combination of three modern methods with an approach similar to that espoused by the vertebrate genome project. The resulting contig N50 of > 1Mb passes proposed assembly standards for both VGP and the Earth BioGenome Project (EBP).

Overall, this is a vast improvement over the current genome assembly which was generated prior to long reads, was good at the time, but has many many gaps.

The annotation looks OK as far as I can tell. To be honest I am beginning to find GO analyses a bit boring, but we don't have anything else, so it is what it is, and hopefully the tools will improve as we go forward.

The expanded and contracted gene families is very nice work, and will likely provide one future avenue of research towards controlling this species. It really pulls the result together and makes it appropriate for this journal. The identification of 154 gene family expansions and 82 contractions is something I would like to see in all genome publications moving forward as it enables phylogenetics to be linked to phenotype and lifestyle.

- I wonder if the high level of gene family expansion and contraction should be noted in the title as it is a really nice result that is likely unique to rapidly adapting species.

- The combination of the DDE transposon gives a mechanism in addition to an observation – very nice.

HSP gene expansion rather than expression change is another really nice observation, and will be well appreciated by a broad audience, especially with the similar result with the ORs. The similar result for the p450s given the pesticide resistance is also fascinating, and provides more targets for RNAi and other possible novel control strategies.

In summary it is an important genome resource that is critically needed, with a wonderful phylogenomic analysis that will move the field forward in the right direction. It is pretty short and well written.

This reviewer would like to see the genome out there as soon as the major issue on access to the data in NCBI is resolved and checked by the editor (see Major issue below)

Thanks for doing some great work!

Major issues:

1. I cannot see the assembly, bioproject or bio sample in NCBI.

I assume this is because the authors asked for an embargo until this manuscript was published.

- Can I please plead that the authors moving forward release the genome assembly as soon as possible after QC pre-publication. This will likely stop others from doing their own genomes – such as the Ag pest 100 group (<http://i5k.github.io/ag100pest>) who are also starting to redo this genome with HiFi technology. Had they known, they would have adjusted their plans, offered to collaborate with you, or chosen a different pest species. So little is being done in Arthropod genomics given the number of species, there is plenty of room to collaborate. Given the pre-prints

are also available I think you are safe to make sequences available on submission rather than acceptance these days.

-

- The best way to solve rapidly the problem that I cannot see the assembly and check it has the same stats as described in the paper is for the editor to check this as a pre-requisite to publication.

- i.e. This will be a major revision, which will be resolved when the editor confirms the sequences are publicly available in NCBI. It will not need confirmation from me – the editor can confirm and then OK for publication.

Minor issues:

Line 69: single-molecular should be single-molecule.

You mentioned busco but I couldn't find the busco results in the supp data. – perhaps I missed it

The HiC contact map (Fig S2) looks a little washed out – perhaps the color scale could be adjusted to better differentiate between more and fewer contacts.

Reviewer #2 (Remarks to the Author):

Jiang et al. Review

Summary: This manuscript by Jiang et al. describes the use of PacBio long reads and Hi-C scaffolding to produce a highly contiguous and complete reference genome assembly for an important insect pest, *Bactrocera dorsalis*. Data presented by the authors indicates that the assembly is high quality, with 2,379 contigs that total 542Mb, only 20Mb larger than the expected genome size of *B. dorsalis* (based on kmer counts). The core conserved eukaryotic gene set is more than 96% complete (but see below), and at least half of the contigs produced from the PacBio assembly can be ordered and oriented using Hi-C (~467Mb or 86% of the total PacBio assembly length using Supplementary Tables 1-2 and 1-3). The authors then use the assembly to identify species-specific qualities of the *B. dorsalis* genome which they believe could significantly contribute to the adaptability and invasiveness of this important pest. Most analyses to identify these species-specific qualities are standard and use previously published software. Jiang et al. also re-analyze previously published RNA-seq data to reinforce the importance of some gene families to adaptability of *B. dorsalis*. Overall, I believe that a publication on this *B. dorsalis* genome has a place in the published scientific literature and will be useful for future genetic studies of the pest, including for development of potential management biotechnologies. Yet when I view the manuscript in light of the journal's scope - "publishes significant advances bringing new biological insight to a specialized area of research" - I'm not certain that I feel it meets the criteria of "new biological insight". There are also some methods that need to be clarified or perhaps even modified entirely. My comments on this are delineated below.

Major comments:

1) The framing of the introduction is entirely focused on the invasiveness and adaptability of *B. dorsalis*. Only the final two sentences really touch on the genome itself, and why it might be useful. But the idea that the authors will "find the genetic basis underlying its invasiveness and rapid adaptation" (L65) is certainly an overstatement. Instead, if the importance and invasiveness of this pest were consolidated into a single paragraph, and the authors spent more time discussing how this high quality genome assembly could be used in the future by the research community - both from a basic and applied perspective - it would broaden the applicability of their work.

2) In their analysis of the genome, the authors only put a portion of their findings into a

comparative framework (13 other species). For example, they look at phylogenetic relationships and divergence times (L123-131), shared gene families (L140), and those that were expanded and contracted (L147). Other findings, they suggest may be important to *B. dorsalis* evolution and adaptation, but they do not appear to have made any comparisons with other species, making their importance unclear. For example, the authors describe that ~14% of genes are in tandem duplicates and suggest this is important to the diversity of *B. dorsalis* complex species (L103-105), but they never compare this to other species (within and without "complexes") to justify their conclusion. As another example, they describe the DDE superfamily of transposases as a significant portion of duplicated genes (L110) and conclude that the drastic expansion of the DDE superfamily in *B. dorsalis* potentially contributes to adaptability (L117). It would be nice to see conclusions such as these being made in a comparative framework.

3) One of the major focal gene families for their analysis was HSPs. The authors determined that HSPs were more abundant in their assembly than for other species used in their comparisons and concluded they were potentially important to the adaptability of *B. dorsalis* to varying climates. To reinforce this conclusion, the authors downloaded and re-analyzed the work of other researchers. Strangely, no comparison is made with the published work of these authors, who come to similar conclusions BUT also found 1 gene, HSP23, which was important to survival following *B. dorsalis* exposure to heat. The authors should be clear about the fact that they are re-analyzing published data in their own results section (L173) AND discuss the findings of the researchers who generated the data in light of their own results. The fact that these data were not generated by Jiang et al, and the findings are not different from Gu et al. does detract from the "new biological insight" criteria mentioned above.

4) As they are currently written, the methods do not provide sufficient detail for other scientists who wish to reproduce them. For example, "After filtering low-quality basis of transcriptome raw reads...", but there is no detail on what they mean by low quality (L395). I would suggest that the authors include a greatly expanded methods section in their Supplementary Material, which describes parameter settings for each software used in their data analysis.

Other comments:

L85-86 - CEGMA was deprecated as of 2015. Also, it is unclear whether the % completeness in the text reflects the CEGMA completeness score or BUSCO.

L88 - chromosome-level scaffolds in "the family Tephritidae", not "Tephritidae species"

L94-99 - It is not immediately clear why this description of GO assignments and KEGG pathways is useful in the absence of specifics or much discussion. What useful information about your biological questions can your reader glean from this?

L103 - Does this number of tandem duplicates differ significantly from other insect species? Particularly within the Diptera?

L110 - "syntenic", not "synteny"

L110 - Does the expansion of duplicated DDE transposases in *B. dorsalis* differ from other insect species?

L133 - "*B. dorsalis* contained more genes and gene families" - can this be made more quantitative?

L134-138 - The meaning of these three sentences is unclear.

L171 - It should be clearly indicated that this work is a re-analysis of existing datasets, and the papers for those existing datasets should be cited.

L181-184 - These sentences seem out of place and they need to be linked to the results in the previous lines.

L331 (Genome Annotation) - how were the tandem duplicates identified. Please describe here or in the supplemental.

L397 - Bowtie2 is a DNA read mapper in the tuxedo suite. It does not determine gene expression levels.

RESPONSES TO REVIEWER COMMENTS

Communications Biology – Manuscript ID COMMSBIO-21-0264-T

Manuscript title: Chromosome-level genome assembly of *Bactrocera dorsalis* reveals its adaptation and invasion mechanisms

Dear Editors and Reviewers,

Thank you for providing your insightful and constructive comments and suggestions. Below, we have prepared a point-point response to each comment put forth by you and provided the details of the revisions made to the revised manuscript, citing the page and line numbers. We believe the incorporated changes based on the suggestions have satisfactorily addressed the concerns raised. We hope that the manuscript will now be deemed suitable for publication in *Communications Biology*.

Responses to the comments of Reviewer# 1

Dear Reviewer,

Thank you for providing your insightful comments and suggestions. Below, we have prepared a point-point response to each comment put forth by you.

- 1) I cannot see the assembly, bioproject or bio sample in NCBI. I assume this is because the authors asked for a embargo until this manuscript was published.

Response: Thank you for highlighting this. We have submitted the *Bactrocera dorsalis* genome data to NCBI and will release the data as soon as this manuscript is accepted. Please check the following link:
<https://dataview.ncbi.nlm.nih.gov/object/PRJNA619226?reviewer=276ooamf10vcqsv26jv64gd4t1>

- 2) Line 69: single-molecular should be single-molecule.

Response: Thank you for highlighting this. We have revised “single-molecular” to “single-molecule” (Page 4; Line 77).

- 3) You mentioned busco but I couldn't find the busco results in the supp data. – perhaps I missed it

Response: We apologise for our oversight. We have unintentionally missed uploading it. We have updated the Busco results in the revised manuscript and added the results to Supplementary Table 3. (Page 5; Line 96).

- 4) The HiC contact map (Fig S2) looks a little washed out – perhaps the color scale could be adjusted to better differentiate between more and fewer contacts.

Response: Thank you for highlighting this. We have improved the HiC contact map in the revised Supplementary Figure 2.

Responses to the comments of Reviewer# 2

Dear Reviewer,

Thank you for providing your insightful comments and suggestions. Below, we have prepared a point-point response to each comment put forth by you.

- 1) The framing of the introduction is entirely focused on the invasiveness and adaptability of *B. dorsalis*. Only the final two sentences really touch on the genome itself, and why it might be useful. But the idea that the authors will "find the genetic basis underlying its invasiveness and rapid adaptation" (L65) is certainly an overstatement. Instead, if the importance and invasiveness of this pest were consolidated into a single paragraph, and the authors spent more time discussing how this high quality genome assembly could be used in the future by the research community - both from a basic and applied perspective - it would broaden the applicability of their work.

Response: Thank you for your insightful suggestion. We have rewritten the entire Introduction section (Pages 2–4; Lines 33–73). Following your suggestion, the importance and invasiveness of *B. dorsalis* were consolidated into the first paragraph (Lines 33–47). However, we partly agree with your suggestion to discuss how this high-quality genome assembly could be used in the future by the research community - both from a basic and applied perspective. We think the Introduction section should lay the background of the study to highlight what is already known and how the present study will fill the gap or complement the existing knowledge. Therefore, in the second paragraph, we have described the importance of high-quality genome assembly in the perspective of previous work to set the background for this study and included a hypothesis statement that describes how the genome assembly of *B. dorsalis* would help to understand the molecular mechanisms underlying the rapid adaptation of *B. dorsalis* to new environments (Lines 48–65). Finally, in the last paragraph, a summary and application of this manuscript have been included (Lines 66–73). Moreover, in the conclusions section, we have clearly described how the knowledge generated from this study assembly could be used by the research community in the future.

- 2) In their analysis of the genome, the authors only put a portion of their findings into a comparative framework (13 other species). For example, they look at phylogenetic relationships and divergence times (L123-131), shared gene families (L140), and those that were expanded and contracted (L147). Other findings, they suggest may be important to *B. dorsalis* evolution and adaptation, but they do not appear to have made any comparisons with other species, making their importance unclear. For example, the authors describe that ~14% of genes are in tandem duplicates and suggest this is important to the diversity of *B. dorsalis* complex species (L103-105), but they never compare this to other species (within and without "complexes") to justify their conclusion. As another example, they describe the DDE superfamily of transposases as a significant portion of duplicated genes (L110) and conclude that the drastic expansion of the DDE superfamily in *B. dorsalis* potentially contributes to adaptability (L117). It

would be nice to see conclusions such as these being made in a comparative framework.

Response: Thank you for your constructive comments. We have updated the results of the comparative genomic analysis in the revised manuscript. We compared the tandemly duplicated genes and segmentally duplicated genes between *B. dorsalis* and 13 other economically significant species for gene duplication. The results showed a high prevalence of these genes in *B. dorsalis* next to *D. plexippus* (n=2460, 16.26%) and *B. mori* (n=320; 1.90%). We have added these results to the revised Supplementary Table 6 and described the related contents in the results section (Page 6; Lines 113–123).

We have also performed a comparative analysis between *B. dorsalis* and 13 other species for the DDE superfamily. We have added the results to revised Supplementary Table 6 and updated the results section of the main text (Page 6; Lines 124–129).

- 3) One of the major focal gene families for their analysis was HSPs. The authors determined that HSPs were more abundant in their assembly than for other species used in their comparisons and concluded they were potentially important to the adaptability of *B. dorsalis* to varying climates. To reinforce this conclusion, the authors downloaded and re-analyzed the work of other researchers. Strangely, no comparison is made with the published work of these authors, who come to similar conclusions BUT also found 1 gene, HSP23, which was important to survival following *B. dorsalis* exposure to heat. The authors should be clear about the fact that they are re-analyzing published data in their own results section (L173) AND discuss the findings of the researchers who generated the data in light of their own results. The fact that these data were not generated by Jiang et al, and the findings are not different from Gu et al. does detract from the "new biological insight" criteria mentioned above.

Response: Thank you for this helpful suggestion. We partly agree with your opinion. To reinforce the analysis of the adaptability of *B. dorsalis* to varying climates, we downloaded and re-analysed some published transcriptome data (SRP141127 and SRP158095) of other researchers using the genome generated in our study as reference. However, for SRP141127, no paper has been published until now; for SRP158095 reported by Gu et al., we have cited the paper on Page 20; Line 429 of the revised manuscript. However, our findings are inconsistent with the previously published data, which could be explained by the following points:

Firstly, we used different methods than those used by Gu et al. for data analyses. Gu et al. performed *de novo* transcriptome assemblies, whereas we have used the genome we sequenced as a reference for the transcriptome analyses.

Secondly, there is only a weak overlap in transcripts responsive to varying climates between our study and that of Gu et al. **The transcriptome analyses in this study displayed no significant differences in *B. dorsalis* Hsp expression levels between varying climates, but *Hsp* genes in *B. dorsalis* tended to be upregulated at a hardening temperature of 38°C in Gu et al.'s research.** We have updated the results of the comparative analysis of the expression Hsps under different environmental conditions in our manuscript (Page 9–10; Lines 196–204). Moreover, the discrepancies observed could also be attributed to the different computational protocols and parameters

used in data analyses. It has been shown that different data analysis approaches could obtain the different transcripts responsive to the same treatments. For example, Widana Gamage SMK et al. have reported similar results [Widana et al. Transcriptome-wide responses of adult melon thrips (*Thrips palmi*) associated with capsicum chlorosis virus infection. PLoS ONE, 2018, 13(12): e0208538.]

Thirdly, **a new and interesting finding in our study was that the expressions of the Hsps transcripts in *B. dorsalis* were higher than that obtained for all transcripts (the median FPKM - 3.07 and 1.68, at 25 °C, 2.94 and 1.64 at 38 °C; 3.90 and 2.86, under fed conditions, and 4.06 and 2.80, under starved conditions, respectively), and the expression levels of Hsps under different conditions (25 °C vs 38 °C and fed vs starved) were not significantly different.** Taken together, these results suggested the intrinsic adaptation system of *B. dorsalis* to high temperatures.

- 4) As they are currently written, the methods do not provide sufficient detail for other scientists who wish to reproduce them. For example, "After filtering low-quality basis of transcriptome raw reads...", but there is no detail on what they mean by low quality (L395). I would suggest that the authors include a greatly expanded methods section in their Supplementary Material, which describes parameter settings for each software used in their data analysis.

Response: Thank you for highlighting this. We have expanded the methods section and added the parameter details for each software used in our data analyses. Please see the revised methods section. The changes were accommodated in the revised methods, and they did not require to be added as a supplementary file.

- 5) L85-86 - CEGMA was deprecated as of 2015. Also, it is unclear whether the % completeness in the text reflects the CEGMA completeness score or BUSCO.

Response: Thank you for highlighting this. We apologise for our oversight. Unfortunately, we have unintentionally missed updating the data. In the revised version, we have added the % completeness (99.78% and 96.96%) to the text (Page 5; Lines 93–97) and updated the CEGMA and BUSCO results in revised Supplementary Table 3.

About the reliability of the % completeness generated by CEGMA, we used both CEGMA and BUSCO to evaluate the integrity of the assembled genome, and results showed that a high degree of completeness both by CEGMA (99.78%) and BUSCO (96.96%), suggesting that the results obtained in this study are reliable.

In addition, there are recent studies on genomes that have used CEGMA to evaluate the integrity of the assembled genome. For example, "Weinstein DJ et al. The genome of a subterrestrial nematode reveals adaptations to heat. *Nature Communications*, 2019, 10: 5268.", "Wu N et al. Fall webworm genomes yield insights into rapid adaptation of invasive species. *Nature Ecology and Evolution*, 2019, 3: 105-115.", "Li S, et al. The genomic and functional landscapes of developmental plasticity in the American cockroach. *Nature Communications*, 2018, 9: 1008."

- 6) L88 - chromosome-level scaffolds in "the family Tephritidae", not "Tephritidae species"

Response: Thank you for highlighting this. We have rephrased the indicated sentence and hence the phrase. Additionally, we have corrected the error at all places of its appearance. Please see Lines 89, 181, 407, and 411.

- 7) L94-99 - It is not immediately clear why this description of GO assignments and KEGG pathways is useful in the absence of specifics or much discussion. What useful information about your biological questions can your reader glean from this?

Response: Thanks for this important question. We apologise for the lack of sufficient information to understand our intent. It is known that GO assignments and KEGG pathways help classifying genes according to their functions. In this study, GO and KEGG analyses were performed to understand the function of the genes identified in *B. dorsalis*. We have added detailed descriptions of GO and KEGG results to the Results section. Please see Page 5; Lines 103–109.

- 8) L103 - Does this number of tandem duplicates differ significantly from other insect species? Particularly within the Diptera?

Response: Thank you for highlighting this. We have improved the description of the comparative genomic analysis. We compared the tandemly duplicated genes and segmentally duplicated genes between *B. dorsalis* and 13 other species. The results showed a high prevalence of these genes in *B. dorsalis* next to *D. plexippus* (n=2460, 16.26%) and *B. mori* (n=320; 1.90%). We have added these results to the revised Supplementary Table 6 and described the related contents in the results section (Page 6; Lines 113–123). Particularly within Diptera, the number of duplicated genes in *B. dorsalis* was markedly higher than in other members.

- 9) L110 - "syntenic", not "synteny"

Response: Thank you. "synteny" has been revised to "syntenic" (Page 6; Line 124)

- 10) L110 - Does the expansion of duplicated DDE transposases in *B. dorsalis* differ from other insect species?

Response: Thank you for your comment. As described in the responses to one of your earlier comments, we have performed a comparative analysis between *B. dorsalis* and 13 other economically important species for the DDE superfamily. We have added the results to revised Supplementary Table 6 and updated the results section of the main text (Page 6; Lines 124–129).

- 11) L133 - "B. dorsalis contained more genes and gene families" - can this be made more quantitative?

Response: Thank you for this instructive suggestion. We apologise for missing the quantitative data. Following your suggestion, we have added the quantitative description regarding the genes and gene family in the revised manuscript. Please see Pages 7–8; Lines 151–156. However, the quantitative data was already included in Supplementary

Table 7 and Supplementary Fig. 4

12) L134-138 - The meaning of these three sentences is unclear.

Response: Thank you for your comment. We apologise for the lack of clarity. We have modified the indicated sentences to make them clear and logical. Please see Page 8; Lines 156–162.

13) L171 - It should be clearly indicated that this work is a re-analysis of existing datasets, and the papers for those existing datasets should be cited.

Response: Apologies for the confusion. As described earlier, in this study, we have performed comparative analyses between the reported data and our data. For this purpose, the transcriptome data with accession numbers SRP158095 and SRP141127 were downloaded from the NCBI SRA database (<https://www.ncbi.nlm.nih.gov/sra/>). We have cited the study for SRP158095 on Page 20, Line 429, but no study for SRP141127 has been reported yet.

14) L181-184 - These sentences seem out of place and they need to be linked to the results in the previous lines.

Response: Thank you for the suggestion. We apologise for our unclear statements. We have modified the results. We believe that the revised sentences have improved the coherence and established a logical flow of the contents. Please see Page 10; Lines 196–214.

15) L331 (Genome Annotation) - how were the tandem duplicates identified. Please describe here or in the supplemental.

Response: Thank you for the suggestion. The tandem and segment duplicates were detected by MCScanX using the command ‘detect_collinear_tandem_arrays.’ We have added this information to the revised manuscript. Page 17; Lines 375–376.

16) L397 - Bowtie2 is a DNA read mapper in the tuxedo suite. It does not determine gene expression levels.

Response: Thank you for highlighting this. We apologise for the mistake. To correct the mistake, we have rephrased the sentence as: “the independent sequenced samples were mapped to the *B. dorsalis* genome using BOWTIE2 end-to-end algorithm” (Page 20; Lines 436–437).

Reviewers' comments:

Reviewer #1 (Remarks to the Author):

So I am going to apologize upfront, as I am going to be a bit of an asshole. Because of this, I am going to break confidentiality and tell you that I am Stephen Richards, so at least you know who to blame.

I had hoped to just see the assembly and SRA data in NCBI or other appropriate databases, download and check the sizes are roughly what you say it is (i.e. just check the assembly exists and is the right species) and then say please accept for publication.

This is very minimal checking of the primary product of a genome assembly paper.

I was working with the Earth BioGenome Project and tracking the world accumulation of high-quality genome assemblies of eukaryotic species with some scripts that scrape NCBI for the data, so if it isn't in there, or some other INSDC database I won't see it, and perhaps someone else will redo the species not knowing.

i.e. The genome assembly doesn't exist if it isn't in the database.

So I followed your link and could indeed see 3 SRA submissions, 2 for HiC data and one for PB HiFi data, but I don't see a submission for the actual genome assembly. If this were a paper with the genome assembly as a side story that might be acceptable, but the title starts with the words "Chromosome-level genome assembly of *Bactrocera dorsalis*" (see attached screen shot)

Is there something stopping researchers from submitting the assembly to NCBI/EBI/DDBJ if you are in China? (I ask because I have seen other groups of researchers working in China have the same process of only submitting the raw read data to the SRA, and having the actual assembly in some other weird place. For example: "Draft genome of the living fossil *Ginkgo biloba* Guan et al." says:

"Raw and transcriptomic data are available from NCBI bioproject PRJNA307642. Supporting genome assemblies, annotations, supplemental data and custom scripts are hosted in the GigaScience GigaDB repository [30]."

i.e. Ginko - basically it's own phylum doesn't have a genome in an INSDC database, and you have to go to the publication to work out where it is.

Anyway, please submit the genome assembly to NCBI, in the same bioproject as the SRA data, and place the genome assembly accession in the publication - perhaps somewhere around line 88 "here we report the

88 high-quality assembled genome of *B. dorsalis* for the first time (Genome assembly accession XXXXXXXXXX)..... in addition to the usual data availability.

Once this is sorted out I am fine for the paper to be published as soon as possible. If there is some reason why you cannot or are not allowed to do this, then make the genome assembly available somewhere else, and describe how to access it - not as good as an INSDC database, but OK if it has to be that way.

Once again I am sorry to be such an asshole about this, but I think no one will find your assembly if you do not make it easy to find, and if that happens there would have been no point in you doing your hard work.

Response to Reviewers

We apologize for not releasing our genome assembly data. However, as described in our manuscript, the *Bactrocera dorsalis* genome project has been deposited at NCBI GenBank under BioProject Accession PRJNA619226 and BioSample Accession SAMN14492331.

In the revised manuscript, we have included the genome data, available at NCBI by Genbank number JABETM000000000, or at:

<https://www.ncbi.nlm.nih.gov/nucleotide/JABETM000000000.1/>.

The Hi-C raw data was also release, which can be accessed using the SRA number SRR15444039 and SRR15254972. We have added this information in the **Data availability** in the revised manuscript.

The RNA-seq data from SRP141127 were obtained from NCBI. Although there is no publication associated with this dataset set, it is available at NCBI <https://www.ncbi.nlm.nih.gov/sra/?term=SRP141127>. This BioProject included 5 items: 3 fed and 2 starved treatments. The samples were all adults and from the Hainan province in China with no special sex requirement.